

# Lower Locus Coeruleus MRI intensity in patients with late-life major depression

Andrés Guinea-Izquierdo[1,2,*], Mónica Giménez[2,*], Ignacio Martínez-Zalacaín[1,2], Inés del Cerro[1,2,3], Pol Canal-Noguer[4,5,6], Gerard Blasco[7], Jordi Gascón[8], Ramon Reñé[8], Inmaculada Rico[8], Angels Camins[7], Carlos Aguilera[7], Mikel Urretavizcaya[1,2,3], Isidre Ferrer[9,10,11], José Manuel Menchón[1,2,3], Virginia Soria[1,2,3] and Carles Soriano-Mas[2,3,12]

[1] Department of Clinical Sciences/School of Medicine, University of Barcelona, Barcelona, Spain
[2] Department of Psychiatry/Bellvitge University Hospital, Bellvitge Biomedical Research Institute (IDIBELL), Hospitalet de Llobregat (Barcelona), Spain
[3] Network Center for Biomedical Research on Mental Health (CIBERSAM), Madrid, Spain
[4] B2SLab/Departament d'Enginyeria de Sistemes, Automàtica i Informàtica Industrial, Universitat Politècnica de Catalunya, Barcelona, Spain
[5] Networking Biomedical Research Centre in the subject area of Bioengineering, Biomaterials and Nanomedicine (CIBER-BBN), Madrid, Spain
[6] Institut de Recerca Pediàtrica, Hospital Sant Joan de Déu, Esplugues de Llobregat (Barcelona), Spain
[7] Imaging Diagnostic Institute (IDI), Bellvitge University Hospital, Hospitalet de Llobregat (Barcelona), Spain
[8] Dementia Diagnostic and Treatment Unit/Department of Neurology, Bellvitge University Hospital, Hospitalet de Llobregat (Barcelona), Spain
[9] Department of Pathology and Experimental Therapeutics/Institute of Neurosciences, University of Barcelona, Hospitalet de Llobregat (Barcelona), Spain
[10] Department of Pathologic Anatomy/Bellvitge University Hospital, Bellvitge Biomedical Research Institute-IDIBELL, Hospitalet de Llobregat (Barcelona), Spain
[11] Network Center for Biomedical Research on Neurodegenerative diseases (CIBERNED), Madrid, Spain
[12] Department of Psychobiology and Methodology in Health Sciences, Universitat Autònoma de Barcelona, Bellaterra (Barcelona), Spain
[*] These authors contributed equally to this work.

Corresponding authors
Virginia Soria, vsoria@bellvitgehospital.cat
Carles Soriano-Mas, csoriano@idibell.cat

## ABSTRACT

**Background.** The locus coeruleus (LC) is the major noradrenergic source in the central nervous system. Structural alterations in the LC contribute to the pathophysiology of different neuropsychiatric disorders, which may increase to a variable extent the likelihood of developing neurodegenerative conditions. The characterization of such alterations may therefore help to predict progression to neurodegenerative disorders. Despite the LC cannot be visualized with conventional magnetic resonance imaging (MRI), specific MRI sequences have been developed to infer its structural integrity.

**Methods.** We quantified LC signal Contrast Ratios (LCCRs) in late-life major depressive disorder (MDD) ($n = 37$, 9 with comorbid aMCI), amnestic Mild Cognitive Impairment (aMCI) ($n = 21$, without comorbid MDD), and healthy controls (HCs) ($n = 31$), and also assessed the putative modulatory effects of comorbidities and other clinical variables.

**Results.** LCCRs were lower in MDD compared to aMCI and HCs. While no effects of aMCI comorbidity were observed, lower LCCRs were specifically observed in patients taking serotonin/norepinephrine reuptake inhibitors (SNRIs).

**Conclusion.** Our results do not support the hypothesis that lower LCCRs characterize the different clinical groups that may eventually develop a neurodegenerative disorder.

Conversely, our results were specifically observed in patients with late-life MDD taking SNRIs. Further research with larger samples is warranted to ascertain whether medication or particular clinical features of patients taking SNRIs are associated with changes in LC neurons.

## INTRODUCTION

The Locus Coeruleus (LC) is a small pontine nucleus with the largest group of noradrenergic (NA) neurons in the central nervous system (CNS) (*Berridge & Waterhouse, 2003*; *Dahlstroem & Fuxe, 1964*). Its ascending and descending projections modulate neuronal activity in numerous targets throughout the CNS (*Loughlin, Foote & Fallon, 1982*; *Loughlin, Foote & Grzanna, 1986*; *Loughlin, Foote & Bloom, 1986*; *Samuels & Szabadi, 2008*). Different studies suggest that alterations in the LC, and, hence, in its projection network, may critically contribute to the onset as well as to the course and symptom profile of a range of neurological and psychiatric disorders (*Betts et al., 2019*; *Charney, 1998*; *Delgado & Moreno, 2000*; *Gannon et al., 2015*; *Leonard, 1997*). Specifically, the LC has been suggested to be one of the initial sites of appearance of pathologically altered tau aggregates in preclinical stages of Alzheimer's disease (AD) (*Braak et al., 2011*; *Braak & Del Tredici, 2015*; *Zarow et al., 2003*), leading to compensatory increases in $\alpha$2A adrenergic receptor levels in regions receiving noradrenergic input (*Andrés-Benito et al., 2017*). Moreover, there is evidence that loss of noradrenergic input exacerbates AD progression (*Marien, Colpaert & Rosenquist, 2004*; *Heneka et al., 2006*; *Grudzien et al., 2007*).

One of the major limitations to investigate the role of the LC in brain disorders with in-vivo non-invasive techniques is that the LC is difficult to visualize with conventional magnetic resonance imaging (MRI) techniques. However, *Sasaki et al. (2006)* identified the LC in vivo as two bilateral hyperintensities adjacent to the floor of the fourth ventricle adapting the parameters of a traditional 2D T1-weighted fast spin echo sequence (for instance, using a higher number of averages and a high in-plane resolution with a large slice thickness). The molecular underpinnings of such hyperintense signal are still unknown (*Betts et al., 2019*), although it was originally thought to be related with the accumulation of neuromelanin in LC neurons.

Neuromelanin is a dark polymer pigment formed as a byproduct of catecholamine metabolism, and it can be found in the LC and substantia nigra pars compacta. The function of this macromolecule is not entirely known, but it has been shown that neuromelanin accumulates throughout life in noradrenergic LC neurons, binding to metals such as iron and copper (*Enochs et al., 1989*; *Enochs et al., 1997*; *Trujillo et al., 2017*). Due to its ability to chelate metals, neuromelanin could be acting as a paramagnetic agent (*Tosk et al., 1992*), and perhaps neuromelanin could be a paramagnetic agent itself (*Shima et al., 1997*). These

properties initially led some authors to hypothesize that neuromelanin could be causing the T1 shortening effects generating the LC contrast in fast spin echo T1 sequences (*Sasaki et al., 2006*).

Moreover, using MRI and post mortem histology, *Keren et al. (2015)* showed that T1-weighted LC hyperintensities co-localized with LC neurons with high neuromelanin concentration. Nevertheless, more recent research suggested that it is unlikely that LC hyperintensities are reflecting neuromelanin accumulation, but rather water content or general neuronal density (*Watanabe et al., 2019*). In this sense, throughout the manuscript, we will avoid using the term neuromelanin-sensitive MRI, although it may be concluded from the literature that LC MRI contrast ratios (LCCRs) seem to be a valid approximation to infer structural integrity of the LC (*Betts et al., 2019*).

There have been different reports of an overall lower LC signal in different neuropsychiatric conditions, such as major depressive disorder (MDD) (*Sasaki et al., 2010*; *Shibata et al., 2008*), schizophrenia (*Shibata et al., 2008*), amnestic mild cognitive impairment (aMCI) (*Takahashi et al., 2015*), and Alzheimer's or Parkinson's disease (*Sasaki et al., 2006*; *Isaias et al., 2016*; *Matsuura et al., 2013*; *Miyoshi et al., 2013*; *Mukai et al., 2013*; *Ohtsuka et al., 2014*), among others (for a review see *Liu et al., 2017*). It is not clear, however, to what extent such alterations show diagnostic specificity, which might be useful for clinical purposes (e.g., differential diagnosis, treatment selection, or prediction of clinical course). This could be important for disorders with partially overlapping clinical manifestations, such as late-life MDD and aMCI, two conditions typically appearing between the seventh and the eighth decades of life (*Panza et al., 2010*; *Steffens, 2012*).

People suffering from late-life MDD or aMCI have an increased risk for the later development of AD. Nevertheless, while aMCI is a common prodromal stage of AD, with a progression rate to AD estimated between 6% to 25% per year (*Petersen et al., 2001*) (for a review see *Hermida et al., 2012*), late-life MDD is not that robustly associated with dementia onset, and results have been more heterogeneous. Overall, research suggests that people suffering from late-life MDD have an approximately twofold increased risk of developing AD (*Byers & Yaffe, 2011*), although it is not clear whether depression is only a risk factor or may be considered a prodromal symptom of dementia (*Byers & Yaffe, 2011*). Interestingly, moreover, co-occurrence of aMCI and depression seems to increase even more the probability to develop to AD, suggesting that both conditions have an additive or synergic effect (*Modrego & Ferrandez, 2004*; *Gabryelewicz et al., 2007*). In this context, it is not clear whether LC MRI intensity discriminates between these disorders and their comorbidity, and, therefore, to what extent this measurement could contribute to identify subjects at a higher risk of developing a neurodegenerative disorder.

In this study, we aimed at comparing LC MRI intensity between patients with late-life MDD and individuals with aMCI, which, for reference purposes, were also compared to a group of healthy controls (HCs). Moreover, as a secondary aim, we specifically assessed a group of individuals presenting comorbidity between the two diagnoses, and evaluated the modulatory effect on our findings of medication and other clinical variables. According to previous literature, we hypothesized that LC MRI intensity will discriminate between

the clinical groups and HCs, and also across patients with late-life MDD, aMCI and a comorbid presentation of both disorders.

## MATERIALS & METHODS

### Participants

The sample consisted of 89 individuals between 60 and 76 years of age. These were divided into three groups: 37 patients with a primary diagnosis of late-life MDD (11 males, mean age [standard deviation, SD]: 68 [4.1] years), 21 subjects with aMCI (9 males, mean age [SD]: 71.5 [2.7] years), and 31 HCs from the same geographical area (11 males, mean age [SD]: 67.7 [4.1] years). Subjects from the clinical groups were consecutively recruited from the Psychiatry and Neurology Departments of Bellvitge University Hospital (Barcelona, Spain), and HCs through advertisements and word-of-mouth. All participants were interviewed with the Mini-International Neuropsychiatric Interview (MINI) (*Sheehan et al., 1998*). Major depressive disorder diagnoses were established by two experienced psychiatrists (VS and MU) according to DSM-IV-TR criteria, and disorder severity was estimated with the Hamilton Depression Rating Scale (HDRS) (*Hamilton, 1960*) and the Geriatric Depression Scale (GDS) (*Sheikh & Yesavage, 1986*; *Yesavage et al., 1982*), which were, however, not used for diagnostic purposes. Amnestic MCI diagnoses were based on a syndromic categorical cognitive staging approach rather than on the use of biomarkers (*Jack et al., 2018*). Diagnosis was established by consensus of two experienced neurologists (JG and RR) and one neuropsychologist (IR) following Petersen criteria (*Petersen, 2004*), including: (1) complaints of memory loss confirmed by informants, (2) objective long-term memory impairments (scores 1.5 SD below mean age and education adjusted normative values in the delayed recall test from the Wechsler Memory Scale III (WMS-III) (*Wechsler, 2004*) and the brief neuropsychological battery NBACE (*Alegret et al., 2012*)), (3) preserved general cognitive functioning (i.e., scores higher than 23 for literate people and higher than 18 for illiterates in the Spanish version of the Mini Mental State Examination (MMSE) (*Folstein, Folstein & McHugh, 1975*)), and (4) intact or mildly impaired daily living abilities (Clinical Dementia Rating (CDR) (*Morris, 1993*) scores of 0.5). All participants were also administered the Vocabulary subtest of the Wechsler Adult Intelligence Scale, Third Edition (WAIS-III) (*Wechsler, 1999*), to estimate the premorbid intelligence quotient (IQ) (*García-Lorenzo et al., 2013*). Importantly, aMCI participants did not present past or current comorbidity with MDD, while nine out of the 37 patients with MDD presented a comorbid aMCI diagnosis. In all cases, medication was not changed and was kept at stable doses for at least one month before MRI acquisition. Fourteen out of 21 aMCI subjects were taking antidepressants for conditions other than MDD, such as impaired psychosocial adjustment, insomnia or headache. Table 1 summarizes the sociodemographic and clinical characteristics of the study sample, while information about antidepressant medication is provided in Table 2.

Exclusion criteria for the study participants included: (1) ages <60 or >76 years, (2) past or current diagnosis of other major psychiatric disorders including substance abuse or dependence (except nicotine), (3) intellectual disability/neurodevelopmental

**Table 1  Characteristics of the study sample.**

| | MDD patients $n = 37$ | aMCI patients $n = 21$ | Controls $n = 31$ | Statistic ($p$ value)[a] df |
|---|---|---|---|---|
| **Sociodemographic characteristics** | | | | |
| Age, years; mean (SD) [range] | 68 (4.1) [60–74] | 71.5 (2.7) [67–76] | 67.7 (4.1) [60–75] | $H = 1.38$ (0.001) 2 |
| Sex, male; n (%) | 11 (29.7) | 9 (42.9) | 11 (35.5) | $X^2 = 1.03$ (0.599) 2 |
| **Clinical characteristics** | | | | |
| HDRS; mean (SD) [range] | 12.1 (7.1) [1–28] | 3.8 (3) [0–8] | 1.3 (2.8) [0–15] | $H = 50.85$ (<0.0001) 2 |
| GDS; mean (SD) [range] | 5.6 (4.3) [0–13] | 2.3 (2.2) [0–9] | 1.23 (2.3) [0–12] | $H = 27.7$ (<0.0001) 2 |
| MMSE; mean (SD) [range][b] | 25.9 (3.3) [13–30] | 25.8 (2.6) [20–29] | 28.9 (1.49) [24–30] | $H = 32.68$ (<0.0001) 2 |
| Long Term Memory[c]; mean (SD) [range] | 5.08 (2.54) [0-10] | 1.95 (1.28) [0–4] | 8.32 (1.72) [6–12] | $H = 53.93$ (<0.0001) 2 |

Notes.

Abbreviations: aMCI, amnestic Mild Cognitive Impairment; df, degrees of freedom; GDS, Geriatric Depression Scale; HDRS, Hamilton Depression Rating Scale; MDD, late-life Major Depression Disorder; MMSE, Mini-Mental State Examination; SD, Standard Deviation.

[a]Statistic value corresponds to Kruskal-Wallis H test for continuous variables and chi-square test for categorical variables.

[b]The cut-off in the Spanish version of the MMSE (*Lobo et al., 1999*) is adjusted for age and schooling years. Scores higher than 23 for literate people and higher than 18 for illiterates indicate preserved general cognitive functioning.

[c]Evaluated with the delayed recall test from the Wechsler Memory Scale III (WMS-III).

**Table 2  Antidepressant treatment at the time of the study.**

| | n (%) [dose-range in mg] | |
|---|---|---|
| | MDD patients $n = 37$ | aMCI patients $n = 21$ |
| *Tricyclics* | | |
| Clomipramine | 2 (5.4) [37.5–75] | 0 (0) [0] |
| Imipramine | 2 (5.4) [75–175] | 0 (0) [0] |
| Amitriptyline | 1 (2.7) [25] | 2 (9.5) [20–25] |
| *Selective serotonin reuptake inhibitors (SSRIs)* | | |
| Sertraline | 1 (2.7) [200] | 0 (0) [0] |
| Citalopram | 2 (5.4) [10] | 4 (19.1) [10–20] |
| Paroxetine | 3 (8.1) [20–30] | 0 (0) [0] |
| *Serotonin–norepinephrine reuptake inhibitors (SNRIs)* | | |
| Venlafaxine | 12 (32.4) [75–300] | 0 (0) [0] |
| Duloxetine | 11 (29.7) [60–120] | 0 (0) [0] |
| Desvenlafaxine | 2 (5.4) [50–100] | 0 (0) [0] |
| *Other antidepressants* | | |
| Mirtazapine | 7 (18.9) [15–30] | 2 (9.5) [7.5–15] |
| Bupropion | 5 (13.5) [150–300] | 0 (0) [0] |
| Vortioxetine | 1 (2.7) [10] | 0 (0) [0] |
| Agomelatine | 2 (5.4) [25] | 0 (0) [0] |
| Trazodone | 4 (10.8) [50–100] | 7 (33.3) [50–100] |

Notes.

Abbreviations: aMCI, amnestic Mild Cognitive Impairment; MDD, late-life Major Depression Disorder; SD, Standard Deviation.

disorders, (4) neurological disorders, (5) Hachinski Ischemic Score >5 to exclude individuals with a high probability of vascular-derived cognitive deficits, (6) presence of dementia according to the DSM-IV-TR criteria and/or a CDR score>1, (7) severe

medical conditions, (8) electroconvulsive therapy in the previous year, (9) conditions preventing neuropsychological assessment or MRI procedures (e.g., blindness, deafness, claustrophobia, pacemakers or cochlear implants), and (10) gross abnormalities in the MRI scan. We also evaluated the presence of vascular pathology in our brain region of interest (pons), finding no significant alterations preventing accurate image processing and analysis.

The study was approved by The Clinical Research Ethics Committee (CEIC) of Bellvitge University Hospital (reference PR156/15, February 17th 2016) and performed in accordance with the ethical standards laid down in the 1964 Declaration of Helsinki and its later amendments (revised in 2013). All participants gave written informed consent to participate in the study.

## MRI scanning protocol

All scans were performed in a 3T Philips Ingenia Scanner (Koninklijke Philips N.V., Netherlands), using a 32-channel head coil. Following previous reports (*Sasaki et al., 2006*), we obtained a modified T1-weighted fast spin-echo sequence of the brainstem for LC visualization (see examples of individual participants in Fig. S1). Acquisition parameters were: TR 600 ms, TE 15 ms, 15 slices; 2.5 mm slice thickness, 0 mm gap, matrix size $404 \times 250$, FOV $170 \times 170$ mm$^2$, acquisition voxel size $0.42 \times 0.68 \times 2.5$ mm$^3$, reconstructed voxel size $0.39 \times 0.39 \times 2.5$ mm$^3$, FA $90^0$, 6 NEX (online averaging), and a total scan time of 15 min. The sections were acquired in the oblique axial direction perpendicular to the floor of the fourth ventricle, covering from the posterior commissure to the inferior border of the pons. Axial T1-weighted turbo-gradient-echo high-resolution whole-brain anatomical images (233 slices, TR = 10.46 ms, TE = 4.79 ms, flip angle = 8°, FOV = $240 \times 240$, $0.75 \times 0.75$ mm isotropic voxels) were additionally obtained for pre-processing purposes and to discard gross structural pathology. Finally, we also obtained a 2D FLAIR sequence in the axial plane (38 slices, TR = 10,000 ms, TE = 140 ms, TI = 2,700 ms, FOV = $230 \times 186$, $0.8 \times 1 \times 3$ mm voxels, 0.6 mm gap) to assess vascular pathology and other potential radiological abnormalities. Throughout the acquisition protocol, we used foam pads and made sure that patients' head was comfortably placed within the head coil to avoid excessive movement.

## Localization and quantification of the LC

To quantify LC MRI intensity, we used a semi-automated ''in-house'' algorithm. We focused our analysis on the region of the dorsal pons. Specifically, our approach was based on previous studies (*García-Lorenzo et al., 2013*) and consisted of two main steps:

*A): Delineation of regions of interest (ROIs):* We defined four rectangular ROIs onto the Montreal Neurological Institute 152 [MNI 152] template (0.5 mm isotropic resolution) using the Statistical Parametric Mapping (SPM 12) software (http://www.fil.ion.ucl.ac.uk/spm/). Specifically, two of these ROIs were symmetrically located in the left and right areas of the dorsal pons where the LC is expected to be found, adjacent to the floor of the fourth ventricle and extending to the level of the inferior colliculi. Importantly, we explicitly avoided including in these ROIs other hyperintense regions, such as the substantia nigra,

to ensure that the LC was the highest signal intensity. The other two ROIs covered the mid portions of the adjacent left and right pontine tegmentum (again, avoiding the inclusion of the substantia nigra), and were used as a reference to standardize LC intensity values (see below). These ROIs, drawn in normalized MNI space, were then denormalized to the native space of each participant. However, due to the small size of the locus coeruleus and the anisotropic voxel size of the images, the LC sensitive sequence was not manipulated and was used as the target image, therefore avoiding interpolation artifacts and preserving the original signal. Consequently, we first linearly coregistered, for each participant and in native space, the whole-brain T1 image to the LC sensitive image. Next, the tissue probability map (TPM) and the four ROIs in MNI space were coregistered (linear coregistration) to the whole-brain T1 image from the previous step, and such individual whole-brain T1 images were non-linearly normalized to the coregistered TPM, which generated a deformation field. The inverse of the deformation field was calculated by using the deformation function in SPM, taking the LC sensitive image as the image to base the inversion on. The four ROIs were then non-linearly denormalized to each participant native space by applying the inverse of the deformation fields with a 4th degree b-spline interpolation. Finally, these ROIs were binarized using a trilinear interpolation and a cut-off value of 0.1 (using the ImCalc function; i1>0.1) in order to better delimitate the ROI area to be extracted; then, each binarized ROI was applied to the LC-sensitive image (ImCalc function; i1.*i2), which effectively removed from these images all the information outside the ROIs. Therefore, the final output of the process were 4 images for participant, in native space, corresponding to the projection of each ROI into the LC-sensitive image. This approach is summarized in Fig. S2. Likewise, Fig. S3 displays the accuracy of the coregistration between the LC, the reference ROI and the brainstem structures in native space.

*B): Quantification of the LC signal with a growing algorithm.* This second step was implemented using an in-house growing algorithm, programmed in MATLAB version 9.3 (R2017b) (The MathWorks Inc, Natick, Massachusetts), which was applied to the final output images from step A) that encompassed the LC region. Specifically, following *García-Lorenzo et al. (2013)*, who defined LC as "the area of 10-connected voxels with the brightest intensity", the current algorithm searched throughout these images the 10 clusters of 10 contiguous voxels (face and/or edge and/or vertex contiguity) with the highest intensity (highest mean intensity of the cluster) (see Fig. 1). These clusters were visually inspected for their anatomical correspondence with the region of the LC. In case the cluster with highest intensity was deemed to be outside the LC region, the second cluster was inspected, and so on. Finally, we obtained, for each participant, the 10-voxels cluster of highest signal intensity within the LC region, as well as the mean intensity value and peak coordinate for this cluster ($S_{LC}$). In agreement with our interest in assessing the intensity of LC, this 10-voxels average represents a good estimation of mean LC intensity and its presumed neural integrity (*Liu et al., 2017*). This is different from a volume measure of the nucleus; also, we did not discriminate intensities across the different territories of the LC (*García-Lorenzo et al., 2013*).

The mean signal intensity from the reference pontine tegmentum cluster was also obtained for each participant ($S_{REF}$). This value was used to control for putative differences

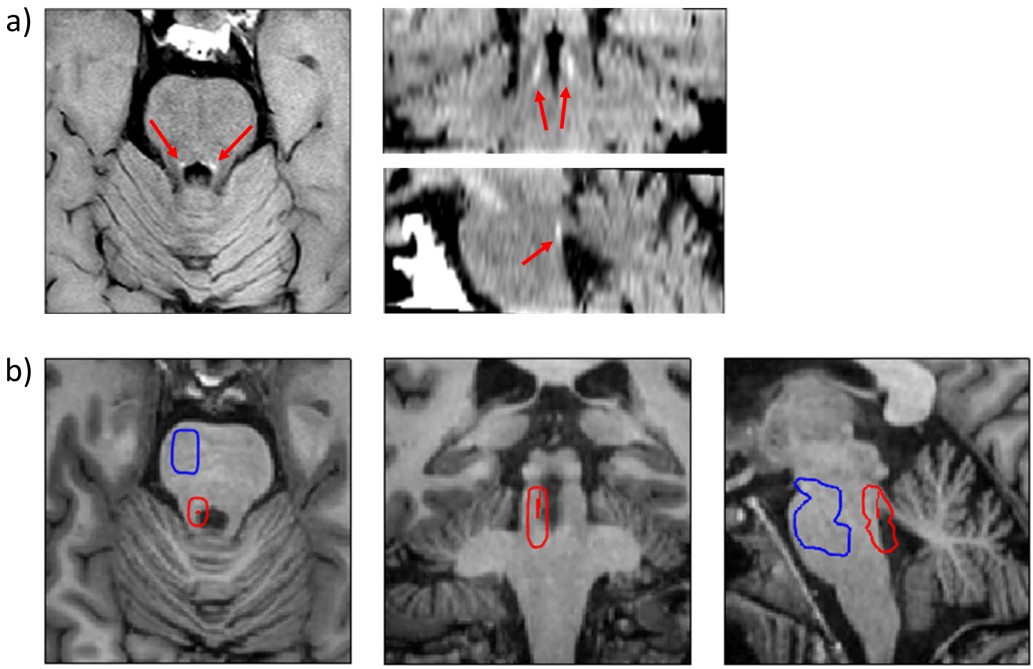

**Figure 1 Visualization and localization of the LC and the reference region.** (A) Axial, coronal and sagittal views of the dorsal pons from the modified fast spin-echo T1 sequence of a representative participant (see text for details). The LC can be identified in each hemisphere as an hyperintense structure (red arrows). (B) Axial, coronal and sagittal views of the dorsal pons from the high-resolution whole-brain T1 sequence of the same participant, depicting, in red, the 10 most intense LC voxels as identified by the growing algorithm. The denormalized ROI of the LC and reference regions are depicted as red (encompassing the 10 most intense LC voxels) and blue contours, respectively.

in global signal intensity across participants. We used the following formula to calculate the LC contrast ratio (CR) (*Liu et al., 2017*): $CR = (S_{LC} - S_{REF}) / S_{REF}$. This CR was calculated for each participant and for both LC (left and right), obtaining two different CRs for each participant ($CR_{Left}$ and $CR_{Right}$). However, as we did not expect interhemispheric differences, and in order to simplify statistical analyses, we averaged left and right CRs using the following formula: *average* LCCR= $(CR_{Left} + CR_{Right}) / 2$. The *average* LCCR was the final variable used in the statistical analysis.

## Statistical analysis

Differences in demographical and clinical features were contrasted across the three study groups with the Kruskal-Wallis test for continuous variables and the Chi-Square test for categorical variables. Differences in *average* LCCR were also assessed across the study groups using the Kruskal-Wallis test. Having in mind that LC MRI contrast ratios interact with age (*Liu et al., 2019*) and that structural and functional differences between men and women have been found in this region (*Bangasser, Wiersielis & Khantsis, 2016*; *Mulvey et al., 2018*), *average* LCCR comparisons were controlled for age and sex. For this, we created a new variable (*standardized residual of average* LCCR, *sra* LCCR) extracting the effect of both

variables, as non-parametrical tests do not allow nuisance covariates. When the Kruskal-Wallis test showed significant differences across the three groups, pairwise comparisons were assessed using the Mann–Whitney's U test. Moreover, we used Spearman's Rho ($r_s$) correlations to evaluate the possible associations between *sra* LCCR and different clinical features (i.e., dose and duration of treatments, MMSE).

Finally, we analyzed the modulatory effect of treatment (e.g., treatment type, duration or dose) on *sra* LCCR with the Kruskal-Wallis test and Spearman correlations (for categorical and continuous variables, respectively). When adequate, statistical significance was adjusted with a Bonferroni correction for multiple testing. We also calculated Cliff's $\delta$ statistic (*Cliff, 1996*) to estimate effect sizes.

## RESULTS

Data on sociodemographic and clinical characteristics of study participants are displayed in Table 1. Due to the consecutive recruitment strategy, the groups differed in age ($H = 13.38$, $p = 0.001$, degrees of freedom (df) $= 2$), with the aMCI group showing the highest mean age, while the MDD group did not differ from HCs. Also, we found a significant negative correlation between the right LCCR and age ($r_s = -0.218$, $p = 0.04$). In any case, as described above, all analyses were controlled for age and sex.

The Kruskal-Wallis test showed significant across-group (MDD $n = 37$, aMCI $n = 21$, HCs $n = 31$) differences in *sra* LCCR ($H = 16.64$, $df = 2$, $p = 0.001$). In post-hoc analyses, it was observed that MDD subjects displayed a lower *sra* LCCR in comparison to aMCI ($U = 193$, $Z = -3.16$, $p < 0.005$, $\delta$ (Cliff's delta effect size) $= -0.53$) and HCs ($U = 281$, $Z = -3.60$, $p < 0.001$, $\delta = -0.51$). See Table 3 and Fig. 2A. Importantly, our groups did not differ in the signal intensity from the reference region ($H = 3.47$, $df = 2$, $p = 0.177$), and, therefore, *sra* LCCR differences were attributable to differences in LC signal intensity.

We did not observe any significant effects of aMCI comorbidity in patients with MDD, since no significant differences were found in *sra* LCCR between the subgroup of MDD without aMCI ($n = 28$) and MDD patients with comorbid aMCI ($n = 9$) ($U = 95$, $Z = -1.10$, $p = 0.272$, $\delta = -0.25$). Likewise, both groups significantly differed from HCs and the aMCI group (see Fig. S4). Moreover, as depicted in Fig. S5, we observed no significant correlations between *sra* LCCR values and MMSE scores for any of the study groups.

When evaluating the possible effects of medication (see Fig. 2B), we observed that patients with MDD taking Serotonin and Norepinephrine Reuptake Inhibitors (SNRIs) ($n = 25$) had a significantly lower *sra* LCCR compared to HCs ($n = 31$) ($U = 143$, $Z = -4.03$, $p < 0.001$, $\delta = -0.63$) and to aMCI patients ($n = 21$) ($U = 94$, $Z = -3.72$, $p < 0.001$, $\delta = -0.64$). However, we did not observe significant differences in *sra* LCCR between MDD patients taking ($n = 25$) and not taking ($n = 12$) SNRIs, although *sra* LCCR of MDDs patients not taking SNRIs did not differ from HCs ($n = 31$) ($U = 138$, $Z = -1.30$, $p = 0.194$, $\delta = -0.26$) or aMCI patients ($n = 21$) ($U = 99$, $Z = -1.01$, $p = 0.312$, $\delta = -0.21$). Although it was not possible to properly evaluate the effect of SNRIs due to the small sample size of the group of MDD subjects exclusively taking SNRIs ($n = 5$), comparing the group of MDD

**Table 3** Average LC Contrast Ratios.

| | Average LC Contrast Ratio Intensity: mean (SD) | Standardized residual of Average LC Contrast Ratio (sra LCCR) Intensity: mean (SD) |
|---|---|---|
| MDD whole group ($n = 37$) | 0.194 (0.059) | −0.450 (0.0728) |
| MDD with aMCI ($n = 9$) | 0.195 (0.084 | −0.474 (1.016) |
| MDD without aMCI ($n = 28$) | 0.194 (0.052) | −0.442 (0.633) |
| aMCI ($n = 21$) | 0.234 (0.079) | 0.257 (0.946) |
| HCs ($n = 31$) | 0.260 (0.093) | 0.363 (1.096) |
| *MDD split according to SNRIs treatment* | | |
| MDD taking SNRIs ($n = 25$) | 0.180 (0.045) | −0.621 (0.538) |
| MDD not taking SNRIs ($n = 12$) | 0.224 (0.076) | −0.094 (0.947) |

**Notes.**
Abbreviations: aMCI, amnestic Mild Cognitive Impairment; HCs, Healthy Controls; LC, Locus Coeruleus; MDD, late-life Major Depressive Disorder; SD, Standard Deviation; SNRIs, Serotonin and Norepinephrine reuptake inhibitors.

patients taking SNRIs alone with the rest of MDD patients ($n = 32$) and with patients taking SNRIs and other psychiatric medications ($n = 20$) we did not find significant differences ($U = 60$, $Z = −0.89$, $p = 0.374$, $\delta = −0.25$; and $U = 43$, $Z = −0.48$, $p = 0.634$, $\delta = −0.14$). Importantly, we did not find any significant differences between MDD patients taking and not taking SNRIs in clinical variables such as severity, disease duration or treatment resistance. These results are presented in Table S1.

Importantly, we repeated the analysis by subdividing the MDD group into those who took ($n = 31$) and those who did not take ($n = 6$) medications with noradrenergic effect (SNRIs + Tricyclics + Mirtazapine + Vortioxetine + Agomelatine + Bupropion), obtaining similar results: MDD taking noradrenergic medication had a significant lower *sra* LCCR compared to HCs ($n = 31$) ($U = 209$, $Z = −3.82$, $p < 0.001$, $\delta = −0.57$) and the aMCI group ($n = 21$) ($U = 143$, $Z = −3.40$, $p < 0.005$, $\delta = −0.56$). Again, although we did not observe significant differences in *sra* LCCR between MDD patients taking and not taking noradrenergic medications, *sra* LCCR of MDD patients not taking noradrenergic medications did not differ from HCs ($n = 31$) ($U = 72$, $Z = −0.86$, $p = 0.387$, $\delta = −0.23$) or aMCI patients ($n = 21$) ($U = 50$, $Z = −0.76$, $p = 0.448$, $\delta = −0.21$). Also, MDD patients exclusively taking noradrenergic medications other than SNRIs ($n = 6$) did not differ from patients taking SNRIs ($n = 25$), not taking noradrenergic medications ($n = 6$), or the HCs and aMCI groups (see Table S2). Finally, we also assessed whether subjects taking adrenergic medication for non-mental health purposes (i.e., beta-blockers; 4 HCs, 1 aMCI and 7 MDD) showed differences in *sra* LCCRs, finding non-significant effects.

Unfortunately, it was not possible to properly evaluate the effect of Selective Serotonin Reuptake Inhibitors (SSRIs) due to the small sample size of the group of MDD subjects exclusively taking SSRIs ($n = 3$). Nevertheless, comparing the group of MDD patients taking SSRIs alone or in combination with other medications ($n = 6$) with HCs and patients with aMCI, we did not find significant differences after Bonferroni correction ($U = 32$, $Z = −2.51$, $p = 0.012$, $\delta = −0.66$, and $U = 20$, $Z = −2.51$, $p = 0.012$, $\delta = −0.68$; Bonferroni p thresholded at $p = 0.0083$).

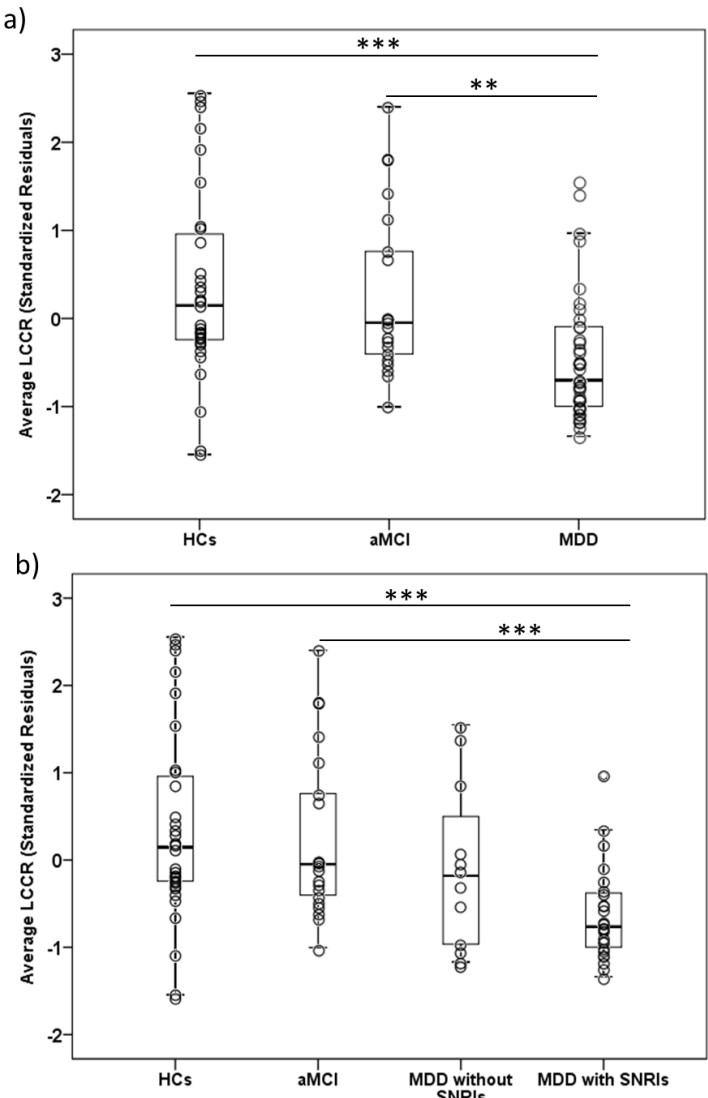

**Figure 2 Boxplots depicting the average LC Contrast Ratios.** Boxplots showing the *average* LCCR (age and sex adjusted, *sra* LCCR) for the different study groups. The individual values (dots) are overlaid for reference. (A) Boxplots of the *average LCCR* for the three main study groups; (B) Boxplots of the *average* LCCR splitting patients with late-life MDD as a function of SNRIs treatment. aMCI, amnestic type Mild Cognitive Impairment; HCs, Healthy Controls; LCCR, Locus Coeruleus Contrast Ratio; MDD, late-life Major Depressive Disorder. ** $p \leq 0.005$, *** $p \leq 0.001$. All results remained significant after excluding outlier values.

Finally, we did not find significant correlations between dose or duration of any of the treatments and *sra* LCCR.

Unless otherwise indicated, all the above significant findings survived Bonferroni correction for multiple comparisons.

## DISCUSSION

Our study compared LC MRI signal intensity between two groups of subjects with brain disorders conferring an increased risk for dementia. We observed that *average* LCCR was significantly lower in late-life MDD patients compared to aMCI and HC groups, but no evidence of additive effects between depression and aMCI. Moreover, such lower LC intensity was associated with the medication used by participants, since significant effects were indeed only observed in the subgroup of MDD patients taking SNRIs.

Our results are in overall agreement with the monoamine hypothesis of depression (*Delgado, 2000*), which suggests that MDD patients show significant alterations in monoaminergic neurotransmission, and, more specifically, with previous neuroimaging studies reporting a lower LC intensity in patients with MDD compared to HCs (*Sasaki et al., 2010*; *Shibata et al., 2008*). Previous reports have shown that NA plays a major role in the pathophysiology of depression (*Delgado & Moreno, 2000*; *Moret & Briley, 2011*), with the LC projecting to emotion- and cognition-related areas implicated in the pathophysiology of depression (*Stahl, 2008*).

In any case, we also observed that this lower LC intensity was specific of MDD patients taking dual action antidepressants (SNRIs). There could be two different explanations for this. On the one hand, if we consider that LC MRI contrast is directly or indirectly associated to the accumulation of neuromelanin compounds (*Betts et al., 2019*), we could speculate about the nature of such accumulation and the molecular effect of dual antidepressant treatment. Neuromelanin is an autophagic byproduct resulting from the metabolism of NA and other catecholamines (*German et al., 1988*; *Graham, 1979*), and its accumulation increases with age in different brain nuclei, such as the substantia nigra and the LC (*Clewett et al., 2016*). Since SNRIs antidepressants inhibit NA reuptake, the levels of this neurotransmitter are increased at the synaptic cleft, what could lead to a decrease in intracellular NA synthesis and metabolism. Sustained over time, this decrease in NA synthesis and degradation could result in a decreased neuromelanin accumulation in the LC. However, this could have been a strong argument if significant correlations between LC signal intensity and SNRIs treatment dose and duration had been observed, but this was not the case in our study. Moreover, there are no evidences of the existence of an enzymatic mechanism capable of inducing degradation or removing neuromelanin (*Fedorow et al., 2006*; *Halliday et al., 2006*), and, therefore, it is not possible to argue that noradrenergic antidepressants may be promoting neuromelanin degradation in LC neurons.

On the other hand, the possibility exists that particular clinical features of the subgroup of MDD subjects who are taking SNRIs might be associated with the lower LCCR in this group of patients. In our sample, MDD patients taking SNRIs did previously fail to respond to treatments with SSRIs. Speculatively, this treatment failure could be related to an impaired NA function reflected in lower MRI LCCR that required the active modulation of NA neurotransmission. Although treatment with dual action antidepressant does not seem to normalize LCCR intensity levels, prospective research on the interaction between antidepressant medication with noradrenergic action and LC signal intensity, carefully controlling for doses and treatment duration, is warranted to elucidate this issue.

Contrary to our initial expectations, we did not observe any significant effect of aMCI on LC signal intensities, which prevents suggesting LC MRI intensity as a putative risk imaging marker for AD. These results contradict previous research showing not only an early loss of LC neurons in AD (*Szot et al., 2006*), but also lower LC intensity values in aMCI versus HCs (*Takahashi et al., 2015*). Critically, however, that study did not exclude patients with comorbid depression, while our group of aMCI individuals did not show comorbidity with major depression. Therefore, we propose that MDD symptoms may be actually more relevantly accounting for lower LC signal intensity. In agreement with this, we have also observed that the presence of comorbid aMCI in patients with MDD did not significantly contribute to further lowering LC signal intensities.

This study is not without limitations. First, since we did not obtain data on cerebrospinal fluid or positron emission tomography imaging biomarkers, we used a syndromic categorical cognitive staging approach and we cannot confirm that our patients were indeed from the AD continuum and will ultimately develop AD. Second, longitudinal studies with larger and carefully characterized samples are highly encouraged in order to better explore the interactions over time of group findings with clinical and sociodemographic variables. This will probably allow identifying clinically relevant subgroups of individuals as a function of their LC signal intensities and contribute to better estimate the likelihood of a degenerative course. Third, we are not using a quantitative MRI method, although other MRI sequences, such as Magnetization Transfer (MT), may provide quantitative indices (*Ramani et al., 2002*; *Sled & Pike, 2001*) that have been used for assessing LC integrity (*Trujillo et al., 2019*). As many other groups, we used a Spin Echo (SE) sequence for assessing LC neuronal integrity (*Liu et al., 2017*), although MT based sequences seem to be a better option to perform absolute signal quantifications (*Trujillo et al., 2019*) and to more precisely localize the LC with isotropic resolution and increased contrast-to-noise ratio, especially at higher-field strengths (*Priovoulos et al., 2018*). We performed a relative quantification taking another brainstem region as the reference area to control for intensity fluctuations across participants, and, importantly, groups did not significantly differ in the intensity of this reference region, indicating that our group differences stem from differences in LC intensity. Nevertheless, further studies combining quantitative and non-quantitative approaches are probably needed to perform better estimations of neural integrity in the LC (*Betts et al., 2019*). Likewise, the present approach did not allow quantifying the volume of the LC, which would have required performing a manual segmentation of the nucleus, and, therefore, we can only infer changes in LC integrity from changes in MRI signal intensity. If such signal intensity is related to neuromelanin accumulation, LC integrity will be probably reflecting preserved adrenergic activity. By contrast, if such signal is related to higher neuronal density or water content, as recently suggested (*Watanabe et al., 2019*), LC integrity will be reflecting a lack of regional atrophy and/or a good osmotic balance in the context of a normal neuronal metabolism.

In the context of limitations, it is also important to note that there are studies suggesting that age-related variance should be considered when exploring structural abnormalities in the LC (*Shibata et al., 2006*; *Liu et al., 2019*). Moreover, these age effects may probably show a non-homogenous distribution along the rostral-caudal axis of the nucleus

(*Betts et al., 2017*; *Dahl et al., 2019*; *Liu et al., 2019*). Thus, age-related declines in LCCR have been mainly observed in the rostral part of the LC in healthy subjects (*Liu et al., 2019*), being such decline in rostral LCCRs related to poorer memory performance in neuropsychological testing (*Dahl et al., 2019*). Although we controlled for age and sex, our algorithm did not allow distinguishing between the rostral and the caudal parts of the nucleus, which may have partially confounded our findings. Moreover, white matter from the pontine tegmentum, used here as reference, has also been shown to increase its T1 intensity with age, probably as a consequence of the highly protracted cycle of myelination of this region, extending up to 70 or 80 years of age (*Yakovlev & Lecours, 1967*). Despite our statistical control for age, and although our groups did not differ in the signal intensity of this reference region, this may have also partially confounded our findings.

## CONCLUSIONS

To sum up, LC signal intensity does not seem to allow identifying subjects with cognitive profiles related to prodromal phases of AD. Conversely, it seems to be associated to late-life MDD. Further research with larger samples should ascertain whether medication aimed at modulating noradrenergic neurotransmission may be playing a role in these findings or, conversely, changes in LC neurons are associated with a particular clinical profile preferentially observed in patients responding to noradrenergic medication. It also remains to be determined if these findings may contribute to optimizing treatment of patients with late-life MDD.

## ACKNOWLEDGEMENTS

The authors are grateful to all the study participants and their families, and to the staff and technicians of Bellvitge University Hospital and Duran i Reynals Hospital who helped to recruit the sample for this study. We thank CERCA Programme/Generalitat de Catalunya for institutional support.

### Funding

This study was supported by the Agency for Management of University and Research Grants of the Catalan Government (2017SGR1247), the Carlos III Health Institute, Spain (Grant PIE14/00034) and FEDER Funds/European Regional Development Fund (ERDF) ('A way to build Europe'). Andrés Guinea-Izquierdo was supported by a FPU14/04822 grant. Ignacio Martínez-Zalacaín is supported by a P-FIS grant (FI17/00294) from the Carlos III Health Institute (Spain). Inés del Cerro is supported by CIBERSAM and previously by a PhD FI Grant from AGAUR-Catalan Government (2016FI_B 00712), grant co-funded by the European Social Fund (ESF) "ESF, Investing in your future". Isidre Ferrer is supported by CIBERNED and CS-M was supported by a Miguel Servet contract from the Carlos III Health Institute (CPII16/00048). The funders had no role in study design, data collection and analysis, decision to publish, or preparation of the manuscript.

## Grant Disclosures

The following grant information was disclosed by the authors:

The Agency for Management of University and Research Grants of the Catalan Government: 2017SGR1247, and 2016FI_B 00712.

The Carlos III Health Institute: PIE14/00034, CPII16/00048, FI17/00294, CIBER and CIBERNED.

FEDER Funds/European Regional Development Fund (ERDF).

FPU14/04822 grant.

The European Social Fund (ESF) "ESF, Investing in your future".

## Competing Interests

These authors report no biomedical financial interests or potential conflicts of interest regarding this work.

Virginia Soria has received grants and served as a consultant or continuing medical education (CME) speaker for Lundbeck, Otsuka, Janssen-Cilag, Exeltis and the Institute of Health Carlos III through the Spanish Ministry of Economy and Competitiveness. Mikel Urretavizcaya has been funded by the Institute of Health Carlos III through the Spanish Ministry of Economy and Competitiveness and has received compensation for lectures, advisories, or grants from Janssen-Cilagand Lundbeck. José Manuel Menchón has received grants and served as a consultant, advisor or CME speaker for Janssen-Cilag, Lundbeck, Medtronic, Otsuka, and the Spanish Ministry of Economy and Competitiveness (CIBERSAM).

The rest of authors have nothing to disclose nor have any financial relationships with commercial interests.

## Author Contributions

- Andrés Guinea-Izquierdo performed the experiments, analyzed the data, prepared figures and/or tables, authored or reviewed drafts of the paper, investigation, and approved the final draft.
- Mónica Giménez analyzed the data, prepared figures and/or tables, authored or reviewed drafts of the paper, and approved the final draft.
- Ignacio Martínez-Zalacaín performed the experiments, analyzed the data, authored or reviewed drafts of the paper, software implementation, and approved the final draft.
- Inés del Cerro performed the experiments, authored or reviewed drafts of the paper, investigation, and approved the final draft.
- Pol Canal-Noguer analyzed the data, authored or reviewed drafts of the paper, software implementation, and approved the final draft.
- Gerard Blasco performed the experiments, authored or reviewed drafts of the paper, and approved the final draft.
- Jordi Gascón, Ramon Reñé, Inmaculada Rico, Angels Camins and Mikel Urretavizcaya performed the experiments, authored or reviewed drafts of the paper, data curation, investigation, and approved the final draft.
- Carlos Aguilera performed the experiments, authored or reviewed drafts of the paper, data curation, funding acquisition, and approved the final draft.

- Isidre Ferrer conceived and designed the experiments, authored or reviewed drafts of the paper, funding acquisition, and approved the final draft.
- José Manuel Menchón conceived and designed the experiments, authored or reviewed drafts of the paper, data curation, funding acquisition, and approved the final draft.
- Virginia Soria conceived and designed the experiments, performed the experiments, authored or reviewed drafts of the paper, data curation, project administration, investigation, and approved the final draft.
- Carles Soriano-Mas conceived and designed the experiments, performed the experiments, authored or reviewed drafts of the paper, data curation, supervision, project administration, funding acquisition, and approved the final draft.

### Human Ethics

The following information was supplied relating to ethical approvals (i.e., approving body and any reference numbers):

The study was approved by The Clinical Research Ethics Committee (CEIC) of Bellvitge University Hospital and performed in accordance with the ethical standards laid down in the 1964 Declaration of Helsinki and its later amendments (revised in 2013) (Reference PR156/15 (17 February 17 2016).

### Data Availability

Raw data is available in the Supplemental Files.

### Supplemental Information

Supplemental information for this article can be found online at http://dx.doi.org/10.7717/peerj.10828#supplemental-information.

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
