# Peer review of "Lower Locus Coeruleus MRI intensity in patients with late-life major depression"

_PeerJ, doi:10.7717/peerj.10828_

## Round 0.1 · original submission · Major Revisions

As you will see, the reviewers were quite thorough and consistent in their recommendations. I encourage you to give careful consideration to their remarks, and I believe that the manuscript will be considerably enhanced if these points are addressed. Thank you again for submitting your work to PeerJ.

Reviewer 1 ·

Basic reporting

Language is mostly clear but more descriptive terms should be used as detailed below. Some suggestions for additional references are given to provide a more exhaustive overview of recent additions to the literature. A few typos are still in the main text but those can be quickly resolved.
* * *
[Major comments]:

- The authors often use a language that suggests within-person (i.e., longitudinal) change in locus coeruleus ratios. For instance, they often use the words “decrease(d)”, “reductions”, “alterations” etc. (including in the title). Their data, however, merely suggests between-person differences. Please adapt your wording throughout the manuscript. That is, based on your cross-sectional data you cannot claim that there is a decrease or reduction.
- Similarly, the contrast mechanism of current locus coeruleus-sensitive MRI sequences is not well understood (see e.g., the Betts et al., 2019 consensus statement in Brain). Thus, please refrain from using terms like “neuromelanin signal” etc. throughout the manuscript.
* * *
[Minor comments]:

- Results: Please always report statistics in full in the main text (not only p-values [e.g., “Due to the consecutive recruitment strategy, groups differed in age (p=0.001) …”]).

- Table 1: Please specify which long-term memory test is reported (in the legend).
- Methods: Did the FSE sequence include online or offline averaging? Please specify.

- Results: Please report whether locus coeruleus ratios were related to age and sex (as corrected values are reported throughout).

- A number of dedicated recent studies demonstrated age differences in locus coeruleus ratios and their topography along the rostrocaudal axis of the nucleus in large samples (https://pubmed.ncbi.nlm.nih.gov/30447418/ ; https://www.nature.com/articles/s41562-019-0715-2 ), please consider adding these to your discussion of (A) age differences (B) rostrocaudal differences to provide a more complete picture of the literature.

- Methods: “However, due to the small size of the locus coeruleus and the anisotropic voxel size of the images, the neuromelanin sensitive sequence was not manipulated and used as reference” – as there is a particular “reference” used to calculate the locus coeruleus ratios, please consider exchanging the word reference here (e.g., use “fixed image”; “target image” instead) as it is misleading.

- Please double-check your reference “German et al., 2988” :)

Experimental design

As detailed below, the methods (MRI-processing) need to be described more thoroughly.
* * *
[Major comments]:

- The description of the coregistration procedure needs more detail so potential readers can reproduce the pipeline. Please add in the main text (A) which software and version was used, (B) what type of coregistration type (linear, nonlinear etc.), (C) what type of interpolation. Related to this:

o The authors appear to threshold (binarize) the transformed ROI images. This suggests that some type of interpolation other than nearest neighbors was used (which would maintain the binarized properties of the ROI image over the transformation). Please adapt/explain.
o As brainstem coregistration is quite challenging for standard software (see e.g., Keren et al., 2009, NIMG or Betts et al., 2019, Brain for discussions), please provide some visualization (in the supplementary information) for coregistration accuracy in the brainstem.

- Methods: Please justify why the mean over the 10 brightest voxels is a valid metric for the (integrity of the) locus coeruleus. Considering the native space voxels size (0.4 * 0.4 * 2.5), 10 voxels do not capture the whole locus coeruleus? Cf. Tona et al., 2017 in Brain Struct. Funct.
Related to this aspect: Please justify why you did not employ the suggested formula to calculate the locus coeruleus intensity ratio, cf. Liu et al. 2017, NBR (Eq. 1), as this limits comparability to other studies.
- Methods: Unless you are expecting interhemispheric differences in the association of locus coeruleus ratios, please consider averaging ratios across hemispheres to (A) yield more reliable integrity estimates (B) reduce the overall number of comparisons (and the need to correct for multiple comparisons).

Validity of the findings
* * *
[Major comments]:
- Methods: What other medication affecting the central noradrenergic systems did participants take (e.g., beta blockers)? Did these relate to locus coeruleus ratios as well?

- Figure 1: Does this visualize a single participant in native space? – the voxel size does not appear to differ substantially between x (y) and z dimensions (should be 0.4 vs 2.5 mm).
* * *
[Minor comments]:

- Figure 1: Please show the same participant with the same sections in panel A and B so potential readers can evaluate the fit of the locus coeruleus and reference ROI to the hyperintensity. Also, please add the reference ROI to the image.

- Table 1: Healthy controls show a MMSE range of 24—30, this indicates that some HCs fall below the often employed cut-off of 26 points? Please report and discuss. In addition, the range of MMSE values indicates that in the MDD group, at least one subject received a score of 13 points? – this seems particularly low?

- Methods: Why was this specific age range (60—76 y) chose, please justify.

Additional comments

General:
In their manuscript titled „Locus coeruleus neuromelanin signal intensity is decreased in patients with late-life major depression“ Guinea-Izquierdo, Mónica Giménez et al. investigate MRI-indexed locus coeruleus integrity in later life. In particular, a group of healthy controls is compared to patients suffering from depression and/or MCI (in groups of n ~ 30). The authors report lower locus coeruleus ratios is depression which may be related to intake of SNRI.
The study is timely and of interest to a broader readership. Locus coeruleus imaging is a relatively new scientific field and the study may help to better understand previously reported age differences in locus coeruleus contrast. However, before publication a series of major concerns need to be addressed. As detailed below/above (depends on the order of these boxes):

- The description and justification of employed methods is often insufficient
- A more descriptive language should be used at several instances
* * *
Reviewer 2 ·

Basic reporting

There are some English language errors:
- Do not use the word “convert” or “conversion” to indicate progression in disease stage. Conversion is a word used in religious context.
- “the more the treatment, the more the LC intensity” : please rephrase as: long treatment was associated greater LC intensity values
- Throughout the manuscript, the authors describe their effects in terms of “increase” or “decrease”. This falsely gives the impression that there is a change over time. Given that the authors are working with cross-sectional data, they should describe the direction of the association (greater, higher, lower).

Reporting:
- Please provide exact p-values, degrees of freedom and effect sizes
- Given that the sample size is so small, the authors should also show the data points on top of the bar plots
- Raw data is supplied. It is not clear whether the authors will share the code as well, as per PeerJ policy

Experimental design

• The T1-weighted TSE sequence used in the current manuscript to structurally image the locus coeruleus differs from standard sequences in terms of spatial resolution1-3. Given the rod-shaped nature of the structure, it would be best to image this structure isotropic or by having an in plane cube and larger slice thickness. Here, the authors choose to use 0.42 * 0.68 * 2.5mm^2, which is then reconstructed to the resolution that is normally used. This may induce partial volume effects coming from the fourth ventricle. The authors should provide images of the LC across various subjects. In addition, the sequence lasts 15 minutes, which is very long for these small voxels and the signal may well be impacted by motion artefacts. How did the authors adjust for motion? Where there individuals were the locus coeruleus could not be visually identified?
• Neuromelanin-sensitive imaging: several quantitative MRI-studies as well as animal studies have now shown that it is unlikely that these contrasts of TSE-scans are reflecting neuromelanin cell density, but rather water content or general neuronal density4. The authors should refrain from the word neuromelanin-sensitive.
• It is know that TSE sequences suffer from intensify variations across the slices, which can cause variability among subjects that is related to the pulse-sequence rather than representing biological variation. How did the authors deal with this?
• Furthermore, previous authors have reported that the pontine tegmentum also exhibits an age-effect and this may confound the calculated integrity measure1. Looking at the dataset provided by the authors, there does seem to be a positive relationship between age and signal in the reference region and not for the intensity of the locus coeruleus. This effect seems to be present mainly in the controls and therefore may have confounded the integrity measure and also the group comparisons. This should be discussed.
• What is the integrity measure referring to? How can it provide information on the function of the LC?
• Did the authors expect a asymmetric pattern in LC intensity? Why not combine left and right into one measure to reduce the number of comparisons?
• The new diagnostic framework for Alzheimer’s disease states that the assessment of the underlying pathology of neurodegenerative diseases required biomarker information. For Alzheimer’s disease this involves the assessment of amyloid and tau pathology5. The authors have no information available on these biomarkers (or did not include it) and as such should be more cautious in their claims that these patients are at risk for AD. Likewise, the control group or even the MDD group may have pathology in their brains, which could explain the lack of differences between aMCI and controls. As such the conclusion that locus coeruleus does not seem to be a biomarker of risk of a neurodegenerative disorders is not supported by the data and should be rephrased.
• The MMSE indicates that the MDD group is cognitively much more impaired than the aMCI group: does this include individuals with dementia?
• The range of depressive symptoms of the MDD groups is 0 to 13 (GDS) or 1 to 28 (HDRS). These individuals are no longer experiencing depressive symptoms. Is that due to medication or is there a broad range in variability in these patients?
• I am confused about the group allocation of medication use. It seems that some patients were taking combinations of different types? How many were only taking SNRIs? How may this effect the results? The authors should also show the group comparisons for the other medication types to fully understand whether this is due to SNRIs or whether this is a spurious observation.
• Exclusion criteria: what is meant with mental disability?
• What do the authors mean with accumulated medication dose? How was this accumulation calculated?

References used in this review
1. Clewett, D.V., et al. Neuromelanin marks the spot: identifying a locus coeruleus biomarker of cognitive reserve in healthy aging. Neurobiol Aging 37, 117-126 (2016).
2. Betts, M.J., et al. Locus coeruleus imaging as a biomarker for noradrenergic dysfunction in neurodegenerative diseases. Brain 142, 2558-2571 (2019), PMC6736046.
3. Priovoulos, N., et al. High-resolution in vivo imaging of human locus coeruleus by magnetization transfer MRI at 3T and 7T. Neuroimage 168, 427-436 (2018).
4. Watanabe, T., Tan, Z., Wang, X., Martinez-Hernandez, A. & Frahm, J. Magnetic resonance imaging of noradrenergic neurons. Brain Struct Funct 224, 1609-1625 (2019), PMC6509075.
5. Jack, C.R., Jr., et al. NIA-AA Research Framework: Toward a biological definition of Alzheimer's disease. Alzheimers Dement 14, 535-562 (2018), PMC5958625.

Validity of the findings

• Statistical analyses: the authors should use non-parametric statistics given that some subsamples are becoming very small in size. There was also no adjustment for multiple comparisons, which may have possibly induced spurious correlations. Given the small sample size, the authors should be encouraged to use equivalence testing (TOST) to confirm lack of an effect.
• In the discussion, there is a lot of speculation on the role of neuromelanin cell density as a possible underlying explanation for the observed effects. Given that several quantitative MRI-studies as well as animal studies have now shown that it is unlikely that these contrasts of TSE-scans are reflecting neuromelanin cell density, but rather water content or general neuronal density4, this should be rephrased and clearly indicated as speculation.

Reviewer 3 ·

Basic reporting

1. How did the authors identify brain damages such as white matter change or microinfarct in the brain without T2 images?
2. The problem of this study is the signal intensity of neuromelanin-sensitive T1 MRI is not absolute quantification of the amount of neuromelanin. So, the authors used the ratio values of LC and reference areas. In order for the reference area to be used as a reference of LC signal change, the reference areas should not be affected by the disease. The authors should provide a valid basis for this point.
3. The statistical values of LC should be controlled for multiple comparison problem.

Experimental design

no comment

Validity of the findings

no comment

Additional comments

This study was tried to evaluate the neuromelanin signal change in the locus coeruleus (LC) in MDD, aMCI, and HC using neuromelanin-sensitive T1 MRI sequences. The authors found decreased LC signal in MDD compared to aMCI and HCs. This study is very interesting in that it is difficult to visualize and quantify LC, and It is also interesting that there is very little research into the role of LC in depression. The neuroimaging methods used in this paper is appropriate.

---

## Round 0.2 · Minor Revisions

As you will see, both reviewers acknowledged the substantive effort and improvement in the revised manuscript, and the remaining suggested changes are readily addressable. I urge you to give careful consideration to the outstanding items, and I believe it is likely that the work will be suitable for publication once these issues are resolved. Thank you again for submitting your work to PeerJ, and I look forward to receiving the revised manuscript.

Reviewer 1 ·

Basic reporting

1. At some instances references should be added to support the claims the authors make (e.g., lines 110; 130, 442 […])

2. In general, the authors may consider adding why an association between the noradrenergic system and AD can be expected. Even though two conditions that are classified as potentially leading to AD are investigated, the mechanistic relation between LC and AD is not introduced.

3. Line 96: What do you mean by the sequences are modified to visualize the LC?


7. Figure 2 / S Figure 4: Is there a duplicated group label?

8. Please add the statistics and table concerning other noradrenergic medication to the supplementary information.

9. Please provide references for the claim that standard MT sequences are quantitative measurements.

10. Please change “T1 signal intensity” to MR (signal) intensity as the basis for the LC contrast is not (only) T1-dependend.

Experimental design

5. Related to my concern raised in the first review: I believe that transforming the ROIs from standard space to native space with a nearest neighbor interpolation would be the appropriate approach that would avoid introducing the additional thresholding (cf. line 260). The authors should consider adapting their procedure.

6. Figure 1 shows that the reference region stretches way more caudally relative to the LC. What was the motivation behind this design?

Validity of the findings

4. As MDD participants show particularly low MMSE scores: Is there a relation between LC ratios and MMSE values? Please provide a scatter plot indicating the correlation with different marker symbols for the subgroups (can be added to supplementary information).

Additional comments

The authors addressed most of my concerns and I find the manuscript much improved. A few remaining remarks are listed below

Reviewer 2 ·

Basic reporting

The authors have done a lot of work to clarify inconsistencies and improve the language.
A few remaining issues:

Line 212: extend should be extent

Discussion: remove "the first study" --> previous work has been done in clinical groups (e.g. Takahashi)

Fig 2B and S4: something went wrong here with the labels?

Please rephrase:"Thus, age-related declines in LCCR have been mainly observed in the rostral part of the LC in healthy subjects (Liu et al., 2019), being such alterations related to disrupted memory (Dahl et al., 2019)." This second part of the sentence is not clear.

S-Figure1: indicate from which diagnostic group the images are taken

Experimental design

I am not convinced by the medication effect, in particular because no difference in MDD patients with or without SNRIs or noradrenergic medication. It seems that the observed differences are simply reflecting the group differences observed earlier. It doesn't seem to be an effect of medication, but rather of group. Are the individuals with SNRIs having worse symptoms, longer disease duration of worse treatment response and isn't that driving the association instead of the type of medication?
In addition (if the above is the case), it would be good to color the dots in box plot in Figure 2B by depression severity
Finally, given the way the groups are made (not pure, but combination of medications) and the sample size, this should also be toned down in the abstract and discussion.

Validity of the findings

Abstract (and conclusion in discussion): the authors need to clearly indicate that these conclusions require more investigation with larger sample sizes.


Discussion: I do not agree that the MT images are only better for absolute quantification of the signal. As previous work has shown MT approaches improve the CNR (Priovoulos et al., 2018; Neuroimage) and at higher field-strength allow for a fast visualisation (with better SNR and isotropic resolution). And in fact, the T1-weighted contrast does contain some MT effect. This should be rephrased.

---

## Round 0.3 · Minor Revisions

The remaining necessary changes are due to the fact that PeerJ does not copy edit articles beyond the level needed for publication formatting. The current manuscript contains a number of typographical and editing errors that might not be picked up by a standard spelling/grammar check. For instance, on p. 7, line 123: "contrast rations" should read "contrast ratios"; p.8, line 134 "such late-life" should read "such as late-life", and in line 177 "Diagnose..." should read "Diagnosis..."

I am confident that this manuscript will be suitable for publication after thorough review and editing for these and similar minor errors. I look forward to receiving the revised manuscript after proofreading, and I thank you again for choosing to submit your work to PeerJ.

---

## Round 0.4 · accepted · Accept

Thank you for the effort and attention you have devoted to the Reviewers' comments and to the editing of the final manuscript. I look forward to seeing the final published version and thank you again for choosing PeerJ as a forum for your research.